# Bedaquiline inhibits the yeast and human mitochondrial ATP synthases

Min Luo[1,4], Wenchang Zhou [2,4], Hiral Patel[3], Anurag P. Srivastava[3], Jindrich Symersky[3], Michał M. Bonar [3], José D. Faraldo-Gómez [2✉], Maofu Liao [1✉] & David M. Mueller [3✉]

Bedaquiline (BDQ, Sirturo) has been approved to treat multidrug resistant forms of *Mycobacterium tuberculosis*. Prior studies suggested that BDQ was a selective inhibitor of the ATP synthase from *M. tuberculosis*. However, Sirturo treatment leads to an increased risk of cardiac arrhythmias and death, raising the concern that this adverse effect results from inhibition at a secondary site. Here we show that BDQ is a potent inhibitor of the yeast and human mitochondrial ATP synthases. Single-particle cryo-EM reveals that the site of BDQ inhibition partially overlaps with that of the inhibitor oligomycin. Molecular dynamics simulations indicate that the binding mode of BDQ to this site is similar to that previously seen for a mycobacterial enzyme, explaining the observed lack of selectivity. We propose that derivatives of BDQ ought to be made to increase its specificity toward the mycobacterial enzyme and thereby reduce the side effects for patients that are treated with Sirturo.

[1] Department of Cell Biology, Harvard Medical School, 250 Longwood Avenue, Boston, MA 02115, USA. [2] Theoretical Molecular Biophysics Laboratory, National Heart, Lung and Blood Institute, National Institutes of Health, 10 Center Drive, Bethesda, MD 20892, USA. [3] Center for Genetic Diseases, Chicago Medical School, Rosalind Franklin University, 3333 Green Bay Rd, North Chicago, IL 60064, USA. [4] These authors contributed equally: Min Luo, Wenchang Zhou. ✉email: jose.faraldo@nih.gov; maofu_liao@hms.harvard.edu; david.mueller@rosalindfranklin.edu

According to the World Health Organization, tuberculosis is one of the top causes of death worldwide with an estimated 1.5 million deaths in 2018[1]. Multidrug and extensively drug resistant forms of tuberculosis are of serious worldwide concern, including in the U.S. Bedaquiline, sold under the trade name Sirturo, is the first drug approved by the FDA in over 40 years as an antibiotic to treat tuberculosis[2,3]. Sirturo was approved as part of a combination therapy to treat adults with multidrug-resistant tuberculosis. However, patients treated with Sirturo have fivefold higher risk of death as compared to the placebo control group (11.4% vs. 2.5%)[4,5]. Indeed, Sirturo carries warnings that administration can cause prolongation of the electrocardiogram QT interval potentially causing fatal heart arrhythmias. Pointedly, reduced mitochondrial function can decrease cardiac activity with associated defects in conduction, including prolongation of the QT interval[6–10].

The $F_1F_o$ ATP synthase (M.W. $\approx$ 600 kDa for the monomer) is highly conserved from bacteria to mammals[11–13]. The ATP synthase comprises a catalytic $F_1$ portion, a peripheral stalk or stator, and the membrane embedded, $F_o$ region. $F_o$ contains a ring of c-subunits (eight in mammals, ten in yeast, 14 in chloroplast, and various stoichiometries in bacteria) that functions as a proton turbine, along with a-subunit that provides the conduits for proton entry and exit[14–18]. The c-ring, connected to the central stalk of $F_1$, rotates driven by the movement of protons down the potential gradient from the cytosol of the cell to the mitochondrial matrix, and the proton flow provides the energy for the phosphorylation of ADP in the active site of $F_1$.

There are selective and nonselective inhibitors of the ATP synthase that bind either to $F_1$ or $F_o$. Oligomycin is an example of an inhibitor that binds to the c-ring of $F_o$[15,19] and is highly selective for the mitochondrial enzyme, while venturicidin is an example of a nonselective inhibitor that likely binds to $F_o$ of both bacterial and mitochondrial ATP synthase[20].

Bedaquiline inhibits ATP synthesis of *Mycobacterium tuberculosis* by binding to $F_o$ of the ATP synthase[21,22]. Biochemical and x-ray crystallographic studies indicate that BDQ binds to the c-ring isolated from *Mycobacterium phlei*[23]. Prior studies have reported that BDQ, while an effective inhibitor of the mycobacterial ATP synthase, has little inhibitory activity against the human enzyme[10,21,24,25]. However, in view of the high incidence of death after the administration of Sirturo, we set out to independently evaluate the effect of BDQ on the mitochondrial ATP synthase. Here we provide biochemical evidence that BDQ inhibits the yeast and human mitochondrial ATP synthase. Using cryo-EM, we identify the binding site for BDQ on the yeast mitochondrial $F_o$, which is highly conserved in the mammalian enzyme. Molecular dynamics simulations are then used to derive a model of the binding pose. These results suggest that inhibition of the human ATP synthase by BDQ might contribute to the high incidence of mortality after its administration and there are likely derivatives of BDQ that can be more selective toward the mycobacterial enzyme.

## Results

### Bedaquiline inhibits the yeast mitochondrial ATP synthase.
We first tested the inhibitory activity of BDQ on the ATPase activity of the yeast $F_1F_o$ ATP synthase. ATP hydrolysis is the reverse of ATP synthesis with the assumption of microscopic reversibility. We first assessed this using highly purified yeast $F_1F_o$ reconstituted into bicelles[26]. As a reference, we assessed the inhibitory properties of oligomycin. The results indicated the BDQ inhibited hydrolysis of the ATP by yeast $F_1F_o$ by nearly 85% with an $IC_{50}$ of about 25 nM (Table 1, Fig. 1a). By comparison, oligomycin inhibited ATP hydrolysis by 85%, but with an $IC_{50}$ of

about 100 nM. Thus, BDQ is a slightly more effective inhibitor than oligomycin of ATP hydrolysis by yeast $F_1F_o$ ATP synthase.

We next measured the $IC_{50}$ of BDQ and oligomycin for the inhibition of ATP hydrolysis using yeast submitochondrial particles (SMP). In this case, the $IC_{50}$ increased to about 1.1 µM for oligomycin and 1.3 µM for BDQ (Table 1). Under these conditions, the inhibition of BDQ is comparable to that of oligomycin.

We also measured the inhibitory effect of BDQ on ATP synthesis of the yeast ATP synthase. The assays were performed using purified mitochondria as this activity is driven by a membrane potential. The mitochondria were able to form a potential gradient as assessed by fluorescence quenching of Rhodamine 123 using ethanol as the substrate for NADH production (Fig. 1d). We measured cyanide sensitive ATP production using a luciferase assay. The results indicated that BDQ inhibited ATP synthesis by yeast mitochondria with an $IC_{50}$ of about 1.1 µM (Table 1, Fig. 1b). By comparison, the $IC_{50}$ for oligomycin was determined to be at about 0.1 µM (Table 1, Fig. 1b). Thus, while exhibiting some variability in the $IC_{50}$ related to the assay, the consistent finding is that BDQ inhibits the mechanisms of both ATP hydrolysis and synthesis of the yeast mitochondrial ATP synthase.

### Bedaquiline inhibits the human ATP synthase.
We set out to determine if BDQ inhibited the human ATP synthase. To this end, we isolated tightly coupled mitochondria from HEK293S cells and converted them to mitoplasts. We used mitoplasts (mitochondria with the outer membrane removed) for the analysis to remove the adenylate kinase present in the intermembrane space. The isolated mitoplasts were able to generate a membrane potential, even after freezing and storage in liquid nitrogen, as determined by fluorescence quenching of Rhodamine (Fig. 1c). Our analysis indicates that BDQ inhibits cyanide sensitive ATP synthesis of the human enzyme nearly completely with an $IC_{50}$ of about 0.66 µM (Table 1, Fig. 1b). In comparison, oligomycin inhibits ATP synthesis nearly completely with an $IC_{50}$ of about 0.005 µM (Supplementary Fig. 1). Thus, BDQ inhibits ATP synthesis by the human ATP synthase.

As a control, we assessed if BDQ inhibited ATP synthesis by a mechanism distinct from binding to the ATP synthase. To determine this, we measured the effect of BDQ on the membrane potential for both yeast and human mitoplasts using the same substrates as used for the ATP synthesis assays. The results indicated that at high concentrations, BDQ had a partial effect on the potential generated by human mitoplasts and yeast mitochondria (Supplementary Fig. 2). This effect, however, was not sufficient to explain the observed inhibition of ATP synthesis by BDQ.

### Single-particle cryo-EM reveals the site of BDQ inhibition.
We used single-particle cryo-EM to identify the binding site for BDQ on the yeast $F_1F_o$ ATP synthase. This analysis was performed essentially as described earlier, using purified enzyme reconstituted into nanodiscs made from palmitoyl-oleoyl-phosphatidylcholine (POPC) and cardiolipin (cardiolipin is 12% by weight)[15]. The cryo-EM structure of the entire $F_1F_o$ ATP synthase was determined to an overall resolution of 3.9 Å, in which $F_o$ is with lower resolution due to flexibility between $F_1$ and $F_o$ (Supplementary Fig. 3). First, we docked the apo structure of $F_1F_o$ (PDB code: 6CP6) into the map of the entire $F_1F_o$ and observed no extra density corresponding to BDQ in $F_1$ indicating BDQ was not bound to $F_1$. To improve the $F_o$ structure, we reprocessed the cryo-EM data starting from particle picking at the center of $F_o$, instead of $F_1F_o$. This strategy generated an improved cryo-EM reconstruction of $F_o$ at an overall resolution of 4.2 Å (Supplementary

| Table 1 Inhibition constants for bedaquiline and oligomycin. | | | | | |
|---|---|---|---|---|---|
| **IC$_{50}$ ($\mu$M) ± SEM** | | | | | |
| **ATP hydrolysis** | | | | **ATP synthesis** | |
| **Purified yeast $F_1F_o$** | | **Yeast SMP** | | **Yeast mitochondria** | |
| **BDQ** | **Oligomycin A** | **BDQ** | **Oligomycin A** | **BDQ** | **Oligomycin A** |
| 0.027 ± 0.0025 | 0.107 ± 0.0011 | 1.28 ± 0.25 | 1.1 ± 0.04 | 1.1 ± 0.07 | 0.1 ± 0.008 |
| | | | | **Human mitoplasts** | |
| | | | | 0.66 ± 0.05 | 0.0054 ± 0.0006 |

IC$_{50}$ for ATP hydrolysis and ATP synthesis for BDQ and oligomycin on yeast and human ATP synthase. Cyanide sensitive ATP synthesis for yeast mitochondria was 51.3 ± 2.6 nmoles/ATP/min/mg protein and was inhibited 98% with oligomycin (1 $\mu$M). Cyanide sensitive ATP synthesis for human mitoplasts was 9.0 ± 0.6 nmoles/ATP/min/mg protein and was inhibited 98% with oligomycin (1.25 $\mu$M). Total ATP synthesis for yeast mitochondria was inhibited by 98.6% with 1.5 mM sodium cyanide and by 59.2% with human mitoplasts.

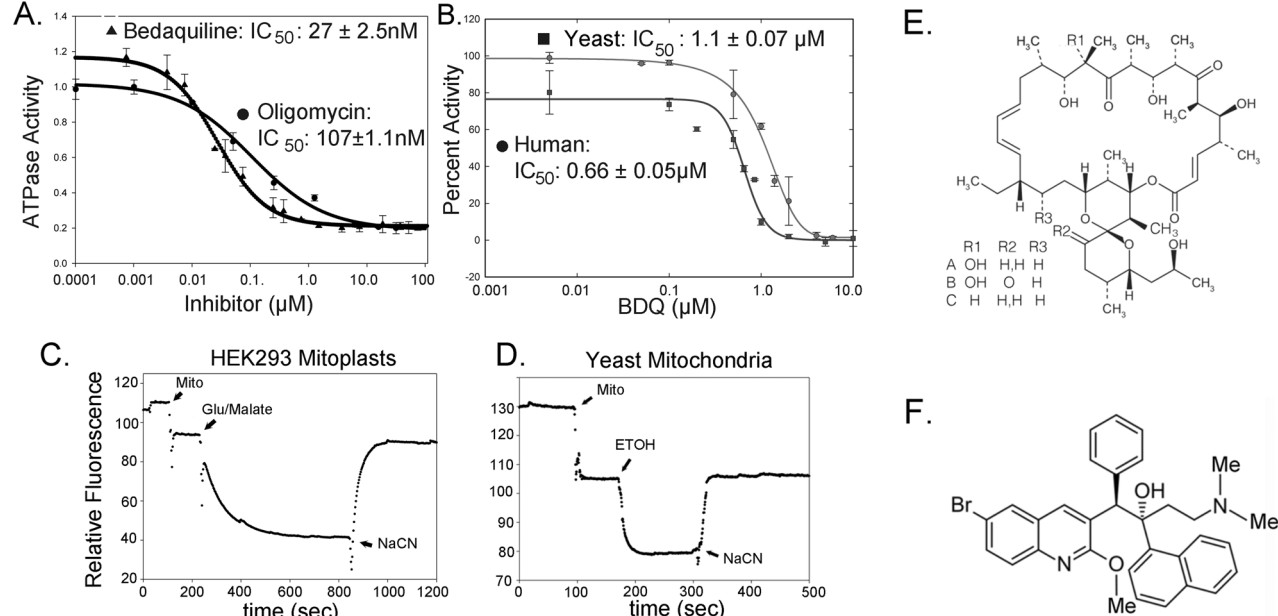

**Fig. 1 Inhibition of $F_1F_o$ ATP synthase. a** Inhibition of the ATPase activity of the purified yeast ATP synthase reconstituted in bicelles. Inhibition of the ATPase activity by oligomycin A (EC$_{50}$: 107 ± 1.1 nM, ($n = 3$)) and BDQ (EC$_{50}$: 27 ± 2.5 nM, ($n = 3$)) **b** Inhibition of the ATP synthase activity by bedaquiline of yeast mitochondria (IC$_{50}$: 1.1 ± 0.05 $\mu$M ($n = 3$)) and human mitoplasts (IC$_{50}$: 0.66 ± 0.05 $\mu$M ($n = 5$)). **c, d** Mitochondrial membrane potential. **c** Human mitoplasts and **d** yeast mitochondria. **e** Structure of oligomycin A, B, and C. **f** Structure of bedaquiline. Membrane potential was measured by fluorescence quenching of Rhodamine 123 (excitation at 488 nm and emission at 525 nm). Rhodamine 123 (1.0 $\mu$M), mitochondria or mitoplasts (150 $\mu$g, mito), ethanol (68 mM, ETOH), glutamate/malate (5.0 mM/2.5 mM), NaCN (0.5 mM).

Fig. 4), which is of good quality with side-chain densities visible for many residues. A unique and strong cryo-EM density is clearly observed at the interface between the a- and c-subunits (Fig. 2, Supplementary Fig. 5, Supplementary Movie 1), whose size and shape are consistent with those of a BDQ molecule. While the resolution is not sufficient to determine unambiguously the exact orientation of BDQ, it is worth noting that this binding site to the c-subunit overlaps with the binding site of oligomycin, a highly specific inhibitor of the mitochondrial ATP synthase[15], supporting that this cryo-EM density reflects bound BDQ.

The presence of BDQ at this site is further supported by changes in the conformation of subunits-a and -c in the region around the proposed binding site (Fig. 3). We compared the structure of the $F_o$ with BDQ bound with that to the structure of the native, oligomycin bound, or the structure of $F_o$ alone, hereafter referred to as the "reference structures"[14,15]. The binding of BDQ disrupted the interface between the a- and c-subunits as compared to the reference structures. Helices 5 and 6 of subunit-a are distorted around the proposed binding site of

BDQ which increases the space between subunit-a and the c$_{10}$-ring. In addition, there are examples of residues showing good side-chain density that are displaced, including aVal207, aMet215, aAsn197, and aGlu223 (Fig. 3a). Pointedly, aMet215 moves by about 3 Å from the reference structures. The side chain of aMet215 is on the face of the helix closest to the BDQ molecule, and would clash with the EM density of BDQ if it was in the reference conformation. Residue aGlu223 is displaced by as much as 5.5 Å. The EM density of aGlu223 is clear for the current model establishing its position. Interestingly, binding of the natural inhibitor protein, IF1, to the porcine ATP synthase also alters the conformation of helix 6 of subunit-a, despite the fact that IF1 binds to $F_1$, distant from $F_o$[27].

There are also changes in the conformation in the c-subunit around the putative BDQ binding site (Fig. 3b). The clearest examples are the movement of cPhe55, 64, 70, and Leu71. For instance, the phenyl ring of cPhe55 is displaced by as much as 3.1 Å and accompanied by distortion of the α-helix. The rearrangements likely accommodate both the steric clashes and the

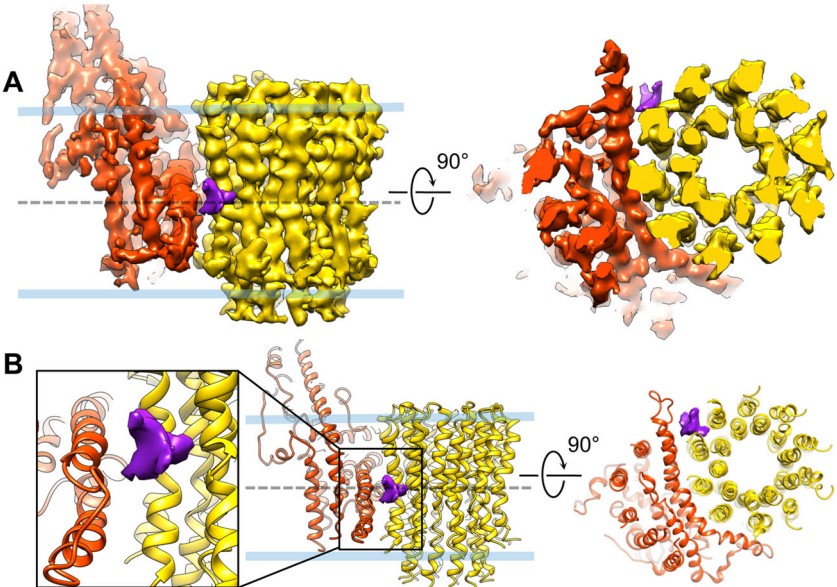

**Fig. 2 Bedaquiline binds in the a/c interface. a** Side (left) and cytosolic (right) views of the cryo-EM maps of $F_o$ with bound BDQ (purple) with the c-ring (gold) and subunit-a (red) contoured at 4.5$\sigma$. **b** Zoomed views of (**a**) with two perpendicular perspectives: a side view and a view from the intermembrane space in the core of the $c_{10}$-ring.

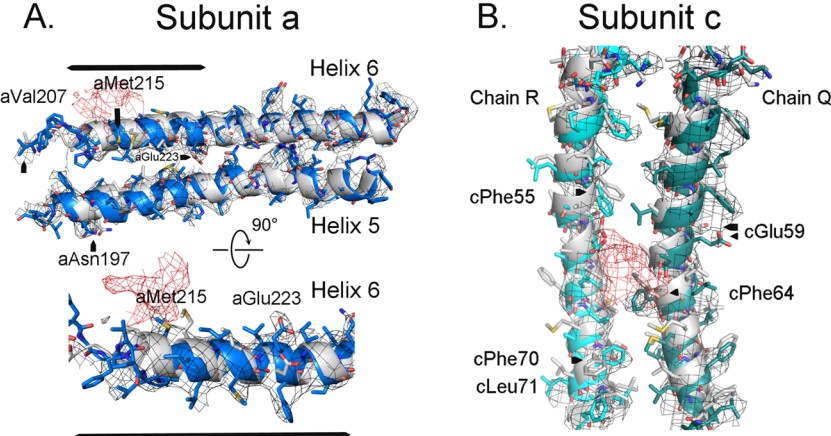

**Fig. 3 Bedaquiline alters the conformation of subunits-a and -c. a** Structural changes in subunit-a (Helices 5 and 6) caused by the binding of BDQ. The reference structure (gray) is the native without anything bound (pdb: 6CP7). **b** Structural changes in subunit-c (chains Q and R) caused by the binding of BDQ. The reference structure (gray) is the native without anything bound (pdb: 6CP7). The cryo-EM density is contoured at 4$\sigma$ with the cryo-EM density attributed to BDQ in red.

favorable interactions with BDQ as in the proposed cation-$\pi$ interaction between cPhe64 and the t-amine of BDQ (see below).

Thus, the distortions in the helices of subunits-a and -c support the conclusion that BDQ is bound at the site identified by the EM density. As mentioned, the binding site for BDQ is similar to that observed for oligomycin[15,19]. A subtle difference, however, is that BDQ appears to form interactions with residues in both subunits-c and -a, while the oligomycin site is confined to subunit-c.

**MD simulations indicate a stable binding pose.** Prior studies have indicated that BDQ binds to the isolated mycobacterial c-ring[23]. To assess if BDQ binds to the mitochondrial c-ring alone, we carried out a series of molecular dynamics simulations, considering two alternative protonation states for BDQ and cGlu59 (Fig. 4a). These simulations showed rapid dissociation if BDQ is deprotonated (neutral) and cGlu59 is protonated (Fig. 4b); by contrast, protonated (ionized) BDQ appeared stably bound when cGlu59 is deprotonated (Fig. 4b), in either of two alternative

poses, referred to as A and B (Fig. 4c). Only the latter pose, however, would be transferable to the a–c interface without major steric clashes. Further simulations of this pose, now in context of the $F_o$ complex (Fig. 4d) showed it to be stable, but only if BDQ is protonated, consistent with the results obtained for the isolated ring (Fig. 4e).

Despite the limited resolution of the cryo-EM data, the predicted mode of binding, depicted in Fig. 4f, recapitulates the main features of the density map. Specifically, when the strongest signals in the cryo-EM map are examined ($\sigma = 6$), a clear density is observed within one of the proton-binding sites in the c-ring, which, according to the MD simulation, originates from the cation-$\pi$ interaction between the t-amine of BDQ and cPhe64 (Fig. 4g).

**Conservation of the BDQ binding site.** The hypothesized mode of interaction (Fig. 4f), whereby the t-amine of BDQ projects into the proton-binding site in the c-ring, happens to be highly similar

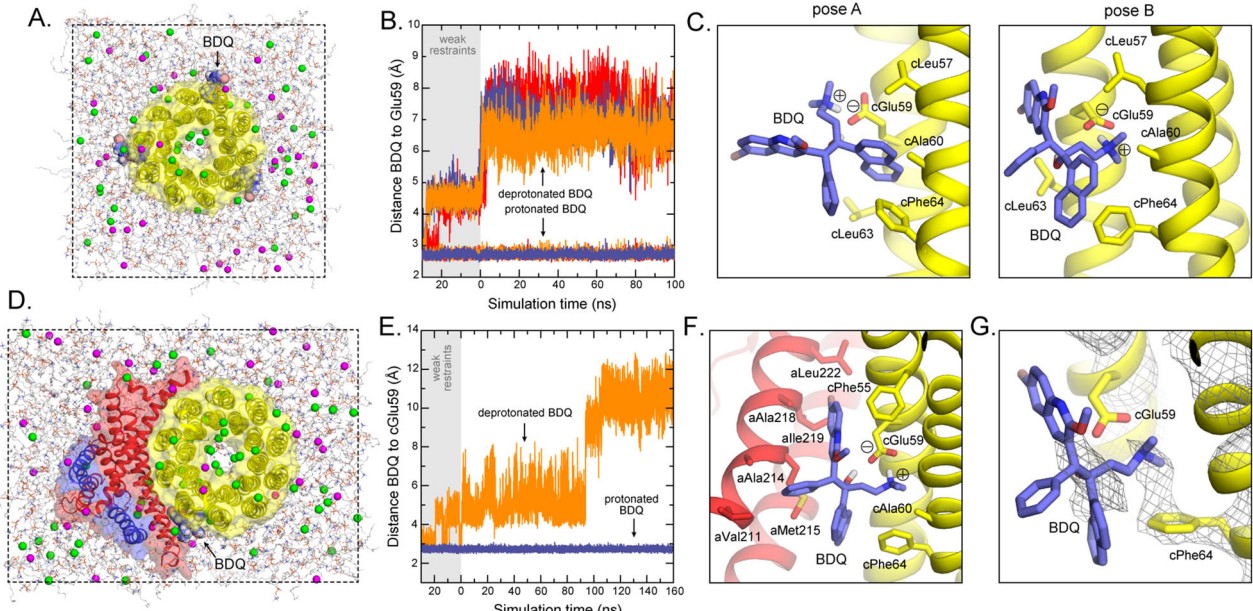

**Fig. 4 MD simulations of bedaquiline bound to the yeast ATP synthase. a** Simulation system, comprising the c-ring (yellow surface/cartoon) embedded in a bilayer of 233POPC lipids and a 100 mM NaCl buffer plus counter-ions (magenta/green). Three bedaquiline molecules (spheres) are initially bound to the c-ring (see "Methods"). The total number of atoms is ~95,000, enclosed in a periodic box of 100 × 100 × 92 Å. Water molecules (ca. 18,000) and lipid hydrogen atoms are omitted for clarity. **b** Time series of the distance between the carboxyl group of cGlu69 and the amine group of BDQ, for each of the three BDQ molecules initially bound to the c-ring. Data are shown for two cases: neutral BDQ (and protonated cGlu59), and protonated BDQ (and ionized cGlu59). For clarity, data are shown only for one 100-ns trajectory; two additional 100-ns trajectories were calculated independently, with the same result. **c** Close-up of the most populated binding modes of protonated BDQ during the simulations (pose A: 44%; pose B: 43%). Neighboring side chains in the protein are highlighted. Nonpolar hydrogen atoms are omitted for clarity. **d** Simulation system, including the c-ring (yellow surface/cartoon), subunit-a (red surface/cartoon) and subunits-b and -8 (blue surface/cartoon) embedded in a bilayer of 254POPC lipids and a 100 mM NaCl buffer, plus counter-ions (magenta/green). One bedaquiline molecule (spheres) is initially bound to the c-ring, at the a–c interface. The total number of atoms is ~109,000, enclosed in a periodic box of 90 × 125 × 101 Å. Water molecules (ca. 20,000) and lipid hydrogen atoms are omitted for clarity. **e** Time series of the distance between the carboxyl group of cGlu69 and the amine group of BDQ, for two cases: neutral BDQ (and protonated cGlu59), and protonated BDQ (and ionized cGlu59). **f** Close-up of the proposed binding mode for protonated BDQ. Neighboring side chains in the both the c-ring (yellow) and subunit-a (red) are highlighted. Nonpolar hydrogen atoms are omitted for clarity. **g** Overlay of the binding pose proposed on the basis of simulation data and the experimental cryo-EM density map, contoured at 6σ.

to that observed for BDQ when bound to the mycobacterial enzyme[23], despite the differences in the amino-acid sequences of these two c-subunits. Side-chain adaptability appears to explain this result. For example, upon BDQ binding to the mycobacterial c-ring, cIle70 stays in place and packs against the BDQ naphthalene ring, while cPhe69 rotates downwards to form T-shape π–π interactions with the other two BDQ rings[23]. In the yeast c-ring, cIle70 is replaced by cPhe64, and so it is this residue that rotates downward to form a T-shape π–π interaction with the BDQ naphthalene ring; cLeu63, which replaces cPhe69, stays in place and packs against the other two rings. This plasticity of the protein side-chain structure, combined with the strong electrostatic interactions between BDQ and the c-ring, appear to underlie the poor selectivity of BDQ for the mycobacterial enzyme.

The cation-π interaction between the t-amine of BDQ and cPhe64 is likely to be a critical interaction that stabilizes BDQ binding. Primary sequence analysis (Supplementary Fig. 7) indicates that cPhe64 is highly conserved from the yeast to mammalian enzyme. Overall, there is a high concentration of aromatic amino acids in and around the binding site and the residues in this region are highly conserved from the yeast to mammalian enzyme. The recently reported cryo-EM structure of the ATP synthase from porcine[27] illustrates this point. Comparison of the structures of the yeast and porcine enzyme shows remarkable conservation at the interface between subunits-a and

-c (Supplementary Fig. 7) including the residue that corresponds to cPhe64. Despite the fact that the c-ring in mammals is composed of eight copies and the yeast c-ring is composed of ten subunits, a side-by-side pair of c-subunits is similar in comparison between them. These results support the conservation of the BDQ binding site identified here from the yeast to the mammalian enzyme.

## Discussion

There were two major surprising results from these studies. First, BDQ inhibited the human enzyme despite reports to the contrary[10,21,24,25]. However, the results from a prior report had suggested that BDQ might inhibit the human enzyme, based on the observation that BDQ has anticancer activity against MCF7 breast cancer cells derived from cancer stem-like cells[28]. Specifically, it was shown that BDQ inhibited mitochondrial oxygen consumption and blocked cell growth with an $IC_{50}$ of about 1 μM. This $IC_{50}$ value is strikingly close to the $IC_{50}$ value we obtained (0.7 μM) for the inhibition of ATP synthesis with mitoplasts derived from HEK293S cells. Second, we predicted that BDQ would bind to the mitochondrial ATP synthase in a mode similar to that described for the isolated mycobacterial c-ring[23], despite the differences in the amino-acid sequence of these enzymes. However, the binding site of BDQ to the yeast ATP synthase is apparently different from that to the mycobacterial enzyme

(Fig. 3). The binding site in the mitochondrial enzyme is at the a/c interface, while all evidence currently indicates that BDQ binds to the c-ring of the mycobacterial enzyme including surface plasmon resonance studies[29]. However, it is possible that BDQ binds to the a/c interface of the mycobacterial enzyme, as this has not been examined. It is also possible that BDQ binds to the surface of the mitochondrial c-ring, but we do not have any cryo-EM density to suggest this.

We believe that the discrepancies between this and other studies of the inhibition of BDQ of the human enzyme are due to differences in the assays and methods, and in some cases, also due to species and cell-specific effects. This is partly illustrated in our studies as well where we get different $IC_{50}$ values for BDQ using yeast SMP as compared to the purified yeast enzyme reconstituted into lipid bicelles which differ by 50-fold (Table 1). In this case, the sample assay differed by lipid composition, ratio of BDQ/enzyme, and the degree of enzyme homogeneity. Like BDQ, oligomycin binds to a site on the c-ring that is buried in the lipid bilayer[15,19]. Both BDQ and oligomycin are amphipathic and will spontaneously partition into the membrane. The effective concentration of free inhibitor in the bilayer very likely depends on the lipid composition. We have previously observed that the oligomycin sensitivity of the ATP synthase reconstituted into nanodiscs was dependent on the lipid composition of the nanodiscs[15]. Thus, lipid composition is likely one of the critical variables that explains the variation in the results for the inhibition of the human enzyme. This concern is also relevant in vivo, where hydrophobic drugs can accumulate in the cell membrane after prolonged administration. Indeed, it is this effect that may account for toxicity of hydrophobic drugs despite having low affinity for their targets.

The partial effect of BDQ at high concentration on the membrane potential, both in human mitoplasts and yeast mitochondria (Supplementary Fig. 2), might be related to the uncoupling effect of BDQ observed on membranes isolated from mycobacterium[30–32] and E. coli[31]. This uncoupling effect was not observed using porcine mitochondria, however, suggesting that BDQ does not act as a general uncoupler[30]. We have also observed that at high concentrations, venturicidin uncouples the yeast mitochondrial ATP synthase, yet it likely binds to $F_o$ in a manner similar to oligomycin[15,19]. The conformational changes at the interface of subunits-a and -c elicited by the binding of BDQ may be responsible for the partial uncoupling of the enzyme. Indeed, this was proposed as a possible mechanism of uncoupling of the mycobacterial enzyme by BDQ[32].

The cause for the occurrence of prolongation of QT interval upon administration of BDQ has not been firmly determined, but BDQ has been shown to inhibit hERG channel activity[9,33] and mutations in the gene encoding hERG result in arrhythmias[34]. Derivatives of BDQ have been developed that reduce the $IC_{50}$ for hERG from about 1.7 μM to >10 μM, though there was no apparent structure–activity relationship[33]. However, studies on the effectiveness of these derivatives on reducing heart arrhythmias have not been reported. While inhibition of hERG can cause prolongation of the QT interval, we suggest that inhibition of the ATP synthase is a contributing factor in the case of the toxicity of BDQ.

The "threshold hypothesis" suggests that there is a minimal energy demand—a threshold value—for any tissue and only once that threshold is crossed, is there a disease state[35–37]. This threshold can differ between cells or organs and explains why some tissues are more sensitive to a decrease in oxidative phosphorylation than others. This differential need is easily recognized in many cancer cells, where most of the ATP is derived by substrate linked phosphorylation and not oxidative phosphorylation, a phenomenon referred to as the Warburg effect[38]. As such, these cells would be predicted to have a very high threshold and can suffer a large loss in mitochondrial function without cell death. In cell-based assays using human cell lines, the threshold level for cell viability should therefore be considered to properly assess the toxicity of drugs that might inhibit oxidative phosphorylation, including BDQ.

With aging, the cell accumulates mutations in the mitochondrial DNA and this can result in a reduction in the mitochondrial membrane potential. In addition, injury to the tissue or drugs or substances can alter the mitochondrial membrane potential. Thus, the risks associated with taking a medication, such as BDQ, that secondarily reduces mitochondrial function seems likely to be greater in elderly patients or patients taking other drugs that also affect mitochondrial function. Greater risk is also associated with high energy demanding tissues, such as the heart. In this regard, administration of BDQ in some patients could reduce mitochondrial ATP synthesis in the heart to a level that crosses the threshold needed for normal organ function.

Knowledge of the similarities and differences in the modes of binding of BDQ to the mitochondrial versus the mycobacterial enzyme provides an opportunity to develop derivatives of BDQ that are more specific toward the mycobacterial enzyme and thereby decrease the serious side effects.

## Methods

The growth of the yeast, the purification of the enzyme, and the preparation of the nanodiscs were described earlier[15]. TMC207-fumarate was a provided by Janssen Pharmaceutica and from the NIH AIDS Reagent Program. BDQ was dissolved in methanol and aliquoted into microcentrifuge tubes and the methanol was removed under vacuum. The dried BDQ sample was stored desiccated under nitrogen or argon, in a sealed bag at 4 °C. The sample was dissolved either in methanol (ATPase experiments) or DMSO (ATP synthesis experiments) and added to the sample for assay. Any remaining sample was discarded. Oligomycin A (Santa Cruz Biotechnology, Santa Cruz, CA) was dissolved in methanol at a concentration of 1–10 mg/ml and stored at −20 °C. For the inhibition assays, the samples were incubated for 5 min with the inhibitor in the reaction buffer at the stated assay temperature just prior to starting the reaction with substrate.

**Statistics and reproducibility.** The inhibition data were analyzed and the $IC_{50}$ calculated using SigmaPlot 11.0 and errors are reported as standard error. For the experiments using isolated mitoplasts from HEK293 cells, only fresh mitoplasts were used with the exception of a control experiment that tested a new batch of BDQ which used a frozen batch. The final results for BDQ inhibition using mitoplasts from HEK293 cells were based on five separated experiments. For mitochondria, SMP, and $F_1F_o$ incorporated into bicelles, isolated from yeast, the samples were frozen and defrosted just before use. For these experiments, the results are based on three experiments. While we only used one batch of yeast SMP and $F_1F_o$, we performed similar experiments on oligomycin B, C, and 21-hydroxy derivatives, which gave similar results. We also performed experiments (but with less data points) using new batches of BDQ and enzyme preparations to assess any variations due to BDQ or sample preparations. The relative integrity of the mitochondria and mitoplasts was assessed by measuring the membrane potential prior to each experiment. We used four lots of BDQ for the experiments: two from Janssen and two from NIH AIDS program. While we did not do a full investigation, we did test each lot to determine if the lots had comparable activities—which they did.

**Mitochondria isolation and mitoplast preparation.** Yeast mitochondria were isolated from W303-1A yeast strain essentially as described[39] but using lyticase to digest the cell wall (lyticase 0.4 mg/g yeast; 90 min. at room temperature). Expression and purification of lyticase was done as described[40]. The mitochondria were frozen and stored in liquid nitrogen. Human mitochondria were isolated from HEK293S GnTI− (ATCC CRL-3022) cells. Cells were grown to 90% confluence in Dulbecco's Modified Eagle Medium supplemented with fetal bovine serum (10%), L-glutamine (4 mM), glucose (4.5 g/L), sodium pyruvate, in the absence of penicillin and streptomycin. HEK293S GnTI− cells were incubated at 37 °C with 5% $CO_2$. Mitochondria were isolated as described[41].

To prepare mitoplasts from mitochondria isolated from HEK293 cells, the mitochondria (0.6–0.7 ml, 5 mg/ml) were incubated on ice for 20 min., with glyco-diosgenin (0.3 mg/ml, Anatrace, Maumee, OH). To this, ice cold IBc buffer was added (0.2 M sucrose, 10 mM MOPS, 1 mM EGTA, pH 7.4) to final volume of 1.2 ml. The sample is centrifuged at 7000 × g for 10 min at 4 °C and the pellet resuspended in 80% of the initial volume with IBc buffer and the protein concentration is measured.

For freezing, the mitoplasts are centrifuged at $7000 \times g$ for 10 min. and resuspended in freezing buffer[42] (10 mM HEPES-KOH pH 7.7, 300 mM trehalose, 10 mM KCl, 0.1% BSA, 1 mM EDTA, 1 mM EGTA) to 10 mg/ml and frozen and stored in liquid nitrogen.

**Bicelle reconstitution of the yeast ATP synthase**. *Saccharomyces cerevisiae* strain USY006 (MAT α, ade2-1, his3-11-15, leu2-3-112, trp1-1, ura3-52, can1$^R$-100, atp2::LEU2, TRP1::ATP2-his6) was used for the isolation of the $F_1F_o$ ATP synthase. The strain, W303-1A (MATa ade2-1, his3-11,15, leu2-3,112, trp1, ura3-1) was used to isolate coupled yeast mitochondria. The yeast ATP synthase was purified and reconstituted into bicelles as described[43]. Briefly, to a final protein concentration of 12 mg protein/ml, 0.7% (w/v) bicelle (3:1 molar ratio of 1,2-dimyristoyl-sn-glycero-3-phosphocholine:1,2-dihexanoyl-sn-glycero-3-phosphocholine) was added from 40% stock, mixed gently and incubated on ice overnight. Another 0.55% of the same bicelle was added the next day for a total 1.25% bicelle (w/v). This preparation was dispensed into small volumes, quick frozen in liquid nitrogen, and stored in liquid nitrogen.

**ATPase activity assay**. The ATPase activity was measured by the coupled enzyme reaction[43] at 30 °C in a buffer containing 0.25 M sucrose, 50 mM K-HEPES, pH 8.0, 3 mM MgSO$_4$, 2 mM ATP. In the assay, protein concentration of the yeast SMP and purified ATP synthase was about 0.3 mg/ml and 1 µg/ml, respectively. For these experiments, the experiments were repeated three times with the same preparation of SMP of $F_1F_o$. However, we purified $F_1F_o$ on at least five different occasions and did not observe any differences. For inhibition studies, the reaction mix was incubated for 5 min with the inhibitor at 30 °C, without ATP, and then started with the addition of ATP.

**ATP synthase activity assay**. ATP synthesis was determined essentially as described previously[44]. Mitochondria (5 µg) was added to buffer (0.65 M mannitol, 20 mM bis-tris-propane, 2 mM phosphate, 0.36 mM EGTA, pH 6.8) and incubated at room temperature for 2 min after which the inhibitor was added and incubation was continued for an additional 5 min and then succinate/ethanol (5 mM/0.8% v/v, for yeast mitochondria) or glutamate/malate (5 mM/2.5 mM for human mitoplasts), and ADP (0.2 mM) was added to start the reaction (note, the ADP must be treated to reduce contaminating ATP. See below). Di-(adenosine-5′) pentaphosphate (40 µM) was added when assaying human mitoplasts to reduce matrix adenylate kinase activity. Di-(adenosine-5′) pentaphosphate increased the background signal and thus was limited to 40 µM. After 60 min, the reaction was stopped with perchloric acid (3.5%) and EDTA (12.5 mM) and the samples were neutralized to pH 6.5 by the addition of NaOH to 0.5 M, and diluted with water by 50–100-fold. ATP was measured with luciferin/luciferase assay in reaction consisting of 25 mM tricine, pH 7.8, 0.2 mM MgSO$_4$, 0.005 mM EDTA, 1 mM DTT, 0.005 mM NaN$_3$, 1 mM D-luciferin and 0.625 µg/ml luciferase and light was measured using a scintillation counter with the channels wide open. A standard curve for ATP was done for all experiments and all reported measurements were within the range of the ATP concentrations used for the standard curve. In some cases, points of the standard curve were repeated if they were outliers. The standard curve was fitted to a polynomial, which was used to calculate the concentration of ATP. Bedaquiline did not affect the activity of luciferase under these conditions.

Substrate ADP was treated with hexokinase and glucose to reduce the level of contaminating ATP. ADP (100 mM) was dissolved in buffer (1 ml, 20 mM Tris-Cl, 0.24 M glucose, 5.5 mM MgCl$_2$, pH 7.6), hexokinase (100 units) was added, and the mixture was incubated at 25 °C overnight. The reaction was stopped by heating the mixture at 65 °C for 20 min, centrifuged at $14,000 \times g$ for 3 min to remove the precipitated protein. This treatment reduced contaminating ATP by 99%.

For yeast mitochondria, the rate of cyanide sensitive ATP synthesis was $51.3 \pm 2.6$ nmoles/ATP/min/mg protein ($n = 3$), was inhibited 98% with oligomycin (1 µM) ($n = 3$, 1 preparation stored in aliquots in liquid nitrogen). Cyanide insensitive ATP synthesis was 0.1 nmoles/min/mg protein. For human mitochondria, the rate of cyanide sensitive ATP synthesis was $9.0 \pm 0.6$ nmoles/ATP/min/mg protein ($n = 5$), was inhibited nearly completely with oligomycin (1.25 µM) ($n = 5$, 5 different fresh preparations). Total ATP synthesis for yeast mitochondria was inhibited by 98.6% with 1.5 mM sodium cyanide and by 59.2% with human mitoplasts. Cyanide insensitive ATP synthesis was likely due to matrix adenylate kinase activity. We did not observe differences in the results using fresh vs. frozen mitochondria or mitoplasts. While the specific activity for ATP synthesis of mitochondria isolated from a human cancer line was not reported in the earlier study[24], the reference for the method used reports a specific activity of about 15 nmoles/min/mg protein (oligomycin sensitive) using mitochondria isolated from MRC5 fibroblasts[45]. Also, for comparison, the specific activity for ATP synthesis using a different method is 0.27 nmoles/min/mg for membranes from *Mycobacterium bovis* membranes and 0.96 nmoles/min/mg for membranes from *Mycobacterium smegmatis*[46].

**Membrane potential measurements**. The membrane potential was assessed by measuring fluorescence quenching of Rhodamine 123 at 488 nm excitation and 525 nm emission in a buffer containing 0.65 M mannitol, 20 mM bis-tris-propane, 2 mM tris-phosphate, 0.3 mM EGTA, pH 6.8 with Rhodamine 123 dissolved in methanol (1.0 µM). Yeast mitochondria were energized with ethanol (0.4% v/v),

| | $F_o$ with bedaquiline (EMDB-21894) (PDB 6WTD) | | $F_1F_o$ with bedaquiline (EMDB-21895) | |
|---|---|---|---|---|
| **Data collection and processing** | | | | |
| Magnification | 31,000 | 36,000 | 31,000 | 36,000 |
| Voltage (kV) | 300 | 200 | 300 | 200 |
| Electron exposure (e–/Å$^2$) | 41 | 48 | 41 | 48 |
| Defocus range (µm) | 0.8–2.8 | 0.8–3.0 | 0.8–2.8 | 0.8–3.0 |
| Pixel size (Å) | 1.23 | 1.169 | 1.23 | 1.169 |
| Symmetry imposed | C1 | | C1 | |
| Initial particle images (no.) | 343,518 | | 1,063,365 | |
| Final particle images (no.) | 47,169 | | 122,736 | |
| Map resolution (Å) | 4.2 | | 3.9 | |
| FSC threshold | 0.143 | | 0.143 | |
| Map resolution range (Å) | 3.5–4.5 | | 3.0–5.0 | |
| **Refinement** | | | | |
| Initial model used (PDB code) | 6CP7 | | | |
| Model resolution (Å) | 4.2 | | | |
| FSC threshold | 0.143 | | | |
| Model resolution range (Å) | 4.0–4.1 | | | |
| Map sharpening B factor (Å$^2$) | −150 | | | |
| Model composition | | | | |
| Nonhydrogen atoms | 9139 | | | |
| Protein residues | 1233 | | | |
| Ligands | 0 | | | |
| B factors (Å$^2$) | | | | |
| Protein | 106.07 | | | |
| Ligand | – | | | |
| R.M.S. deviations | | | | |
| Bond lengths (Å) | 0.007 | | | |
| Bond angles (°) | 1.154 | | | |
| Validation | | | | |
| MolProbity score | 1.86 | | | |
| Clashscore | 7.39 | | | |
| Poor rotamers (%) | 0.86 | | | |
| Ramachandran plot | | | | |
| Favored (%) | 92.78 | | | |
| Allowed (%) | 7.22 | | | |
| Disallowed (%) | 0.00 | | | |

**Table 2 Cryo-EM data collection, refinement, and validation statistics.**

whereas human mitochondria were energized with glutamate/malate (5 mM/2.5 mM).

**Cryo-electron microscopy data acquisition**. Purified yeast $F_1F_o$ reconstituted in nanodiscs at the concentration of 1 mg/ml (2.5 µl, ≈1.7 µM) was applied to a glow-discharged Quantifoil holey carbon grid (1.2/1.3, 400 mesh), and blotted for 3 s with ~91% humidity before plunge-freezing in liquid ethane using a Cryoplunge 3 System (CP3, Gatan). For cryo-EM[15], incubation of $F_1F_o$ with BDQ was performed in two ways: one is on ice, from which 30 mM stock solution of BDQ in dimethyl sulfoxide was added to a final concentration of 30 µM (twofold higher stoichiometry than of the c-subunit at ten copies/complex), and incubated for 30 min; another one is incubation at room temperature for 10 min, with final concentration of BDQ at 5 µM. The mixture (3.5 µl) from either incubation was applied to a grid, blotted and plunge frozen. Cryo-EM data of sample incubated on ice were recorded on a 300 kV Polara electron microscope (FEI) at Harvard Medical School, data of the sample incubated at room temperature were collected on a 200 kV Talos Arctica microscope (FEI) at University of Massachusetts Medical School. All cryo-EM movies were manually recorded with a K2 Summit direct electron detector (Gatan) in super-resolution counting mode using UCSFImage4[47]. Details of the EM data collection parameters are listed in Table 2.

**EM image processing**. EM data were processed as previously described[15]. Dose-fractionated super-resolution movies collected using the K2 Summit direct electron detector were binned over $2 \times 2$ pixels, and subjected to motion correction using the program MotionCor2. A sum of all frames of each image stack was calculated following a dose-weighting scheme, and used for all image-processing steps except

for defocus determination. CTFFIND4 was used to calculate defocus values of the summed images from all movie frames without dose weighting. Particle picking was performed using a semi-automated procedure with SAMUEL and SamViewer. The images collected from Polara and Talos Arctica were combined together for the final data process. Two- and three-dimensional (2D and 3D) classification and 3D refinement (Supplementary Fig. 5) were carried out using "relion_refine_mpi" in RELION. Masked 3D classification focusing on $F_o$ with residual signal subtraction was performed following a previously described procedure. All refinements followed the gold-standard procedure, in which two-half data sets were refined independently. The overall resolutions (Supplementary Fig. 6) were estimated based on the gold-standard criterion of Fourier shell correlation (FSC) = 0.143. Local resolution variations (Supplementary Fig. 6) were estimated from two-half data maps using ResMap. The amplitude information of the final maps was corrected by applying a negative B factor using the program bfactor.exe.

**Model refinement**. The initial model of $F_o$ was derived from our previous ATP synthase model (PDBs: 6CP7). Initial model was rigid-body fitted to our cryo-EM map, extensively rebuilt in Coot and refined in Refmac using the script *refine_local*, and subsequently, using real-space refinement in Phenix essentially as described earlier[15]. The final model was validated with statistics from Ramachandran plots, MolProbity scores, and EMRinger scores (Table 2). MolProbity and EMRinger scores were calculated as described[15].

**Molecular dynamics simulations**. To guide the interpretation of the experimental cryo-EM data, we carried out a series of molecular dynamics simulations aimed at identifying an energetically favorable binding pose that is also consistent with the density map. As a first step, we simulated BDQ bound to the isolated c-ring, embedded in a POPC lipid bilayer (Fig. 4a). BDQ molecules were modeled on three nonadjacent proton-binding sites, in a tentative initial configuration that appeared compatible with the cryo-EM density data, but differed from that observed previously for the mycobacterial c-ring[23]. These simulations were based on the high-resolution crystal structure of the yeast mitochondrial c-ring (PDB 3U2F)[15]. Two possibilities were considered, for each site: one in which cGlu59 in the c-subunit is protonated while BDQ is neutral, and another in which the amine group in BDQ is protonated while cGlu59 is ionized. In the former case, cGlu59 donates a hydrogen bond to BDQ; in the latter, cGlu59 and BDQ form a salt-bridge. The two systems were equilibrated following a staged protocol comprising a series of restrained simulations. The protocol consists of both positional and conformational restraints, gradually weakened over 90 ns, and individually applied to protein side chains and backbone as well as the BDQ molecule. Subsequently, three simulated trajectories of 100 ns each were calculated for each system, free of any restraints. In the simulation where BDQ is neutral and cGlu59 protonated, the three inhibitor molecules were seen to gradually dissociate from the c-ring, spontaneously (Fig. 4b). By contrast, ionized BDQ was found to be stably bound in simulation, for the three binding sites considered and in three independent calculations (Fig. 4b) (note that this ionized configuration is also the most probable given the much greater proton affinity of a methylated amine compared with a carboxylic group; their solution $pK_a$ values are 10.6 and 4.0, respectively). Analysis of the simulation data for bound BDQ show the inhibitor to be quite dynamic; nonetheless, a classification of the configurations sampled according to pairwise similarity revealed two major binding modes, referred to as "pose A" and "pose B" and shown in Fig. 4c. The primary difference between these two poses is that in pose A both BDQ and cGlu59 project away from center of the binding site, whereas in pose B the inhibitor packs more closely against the protein and cGlu59 is retracted into the site.

We next evaluated whether these poses are compatible with the interaction site revealed by our cryo-EM data. Superposition of pose A onto the cryo-EM structure of the $F_o$ complex revealed evident steric clashes between the inhibitor and the backbone and side chains of TM3 of subunit-a; therefore, this pose was discarded. Pose B, however, appeared to be sterically compatible, and therefore we set out to examine it further through simulations analogous to those described above, now for the $F_o$ complex, again in a POPC membrane (Fig. 4d). These simulations were based on the high-resolution cryo-EM structure of the yeast ATP synthase (PDB 6B2Z)[14]. Specifically, the construct considered is a subcomplex that includes the complete c-ring, subunit-a, and the transmembrane regions of subunit-b and subunit-8, i.e., the construct omits the $F_1$ sector. To ensure that the relative arrangement of the four subunits included is preserved during the simulations, despite the absence $F_1$ sector, a weak RMSD restraint was applied to the transmembrane Cα traces of the four subunits, collectively (residues 2–39, 44–73 for subunit-c; residues 28–47, 57–77, 86–104, 115–145, 155–200, 210–246 of subunit-a; residues 63–87 of subunit-b; and residue 11–39 of subunit-8). The force constant of this restraint ($k = 4$ kcal/mol/Å$^2$) was chosen to impact minimally the conformational variability of the c-ring, relative to what was observed in the unrestrained simulations (average RMSD values of $1.01 \pm 0.06$ Å vs. $1.03 \pm 0.12$ Å, respectively). Otherwise, the $F_o$ complex could tumble freely in the lipid membrane, and all protein side chains as well as the BDQ inhibitor itself were also free to reconfigure. For completeness, we again considered two alternative protonation states for BDQ and cGlu59. These systems were prepared and equilibrated using a protocol analogous to that described above for the simulations of BDQ bound to the c-ring. Trajectories of 160 ns were calculated to evaluate the stability and

binding mode of BDQ for each protonation state. The outcome of these simulations is discussed in "Results" and summarized in Fig. 4e–g.

All simulations were carried out with NAMD2[48] using the CHARMM36 force field[49,50] periodic boundary conditions and constant temperature (298 K) and semi-isotropic pressure (1 atm). A force field for bedaquiline (BDQ) was developed as described below. Long-range electrostatic interactions were calculated using PME, with a real-space cutoff of 12 Å. Van der Waals interactions were computed with a Lennard–Jones potential, cutoff at 12 Å with a smooth switching function taking effect at 10 Å. The protein–ligand complexes were embedded in pre-equilibrated hydrated bilayers using GRIFFIN[51]. Clustering analyses based on pairwise similarity used the method of Daura et al.[52] with an RMSD threshold of 1 Å.

**Electronic structure calculations and force field development for bedaquiline**. Force field parameters for BDQ compatible with CHARMM36 were derived in two steps. First, electronic structure calculations were carried out with Gaussian 09 for bedaquiline and for a derivative where the bromine atom is replaced with chlorine (Supplementary Fig. 6). In both cases, the input structure was extracted from PDB 4V1F[23]. The molecular geometry of each of these two compounds, assumed to be in the protonated form, was optimized using Hartree–Fock theory, initially using the SDD basis set, and subsequently with the 6–311 G** basis set. Natural bonding orbital and natural population analysis were then used to calculate the charge distribution of the geometry-optimized structures, using mPW2PLYP theory and the 6–311 G** basis set. Second, a set of force field parameters was derived for the chlorine derivative of BDQ using the GAAMP server[53] on the basis of the same input structure. Briefly, an initial parameter set was deduced from the CHARMM General Force Field, which includes entries for a broad set of chemical compounds. Parameters deemed to be suboptimal for BDQ were then optimized through ab initio calculations at the Hartree–Fock 6–31 G* level and MD simulations. From here, a force field for bedaquiline was derived by adjusting the atomic charges in the chloride derivative to reflect the differences between the two mPW2PLYP calculations mentioned above. Analogously, partial charges for the neutral form of BDQ were obtained by adjusting the amine group. A final MD simulation of 100 ns was then carried out in water to evaluate the resulting force field, resulting in one primary conformer, Supplementary Fig. 6.

**Reporting summary**. Further information on research design is available in the Nature Research Reporting Summary linked to this article.

## Data availability
Two three-dimensional cryo-EM density maps of the nanodisc-embedded yeast mitochondrial ATP synthase bound with BDQ have been deposited in the Electron Microscopy Data Bank under accession numbers: EMD-21895 ($F_1F_o$) and EMD-21894 ($F_o$). The corresponding atomic coordinates for the atomic models have been deposited in the Protein Data Bank under accession numbers 6WTD ($F_o$). All other relevant data are available from the authors upon request. To obtain data relevant to the biochemistry, contact D.M.M., for the cryo-EM, contact M.L., and for date relevant to computational work, contact J.D.F-G. All source data underlying the graphs and charts presented in the main figures for the biochemistry are available as an Excel file in Supplementary Data 1.

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

## Acknowledgements
TMC207-fumarate (BDQ) was a provided by Dr. Koen Andries of Janssen Pharmaceutica NV, Belgium and from the NIH AIDS Reagent Program. We acknowledge Dr. Chen Xu and Dr. Kangkang Song for their help on data collection at the University of Massachusetts Medical School. The work was supported by a grant from NIH R01GM66223 and R35GM131731 to D.M.M. W.Z. and J.D.F.-G. are funded by the Division of Intramural Research of the National Heart, Lung, and Blood Institute. Thanks to Dr. Jun-Yong Choe (Rosalind Franklin University) for the HEK293S starter culture, to Dr. Ahmad Reza Mehdipour (Max Planck Institute of Biophysics, Frankfurt) for allowing us to compare our forcefield parameters for BDQ with his own, and to Melissa G. Chambers for her assistance in the cryo-EM experiments.

## Author contributions
M. Luo designed and performed EM data collection and image processing, and helped in the writing, H.P. and A.P.S. designed and performed experiments and helped in the writing, W.Z. performed the computer simulations and ab initio calculations, analyzed data, and helped in the writing, J.S. helped in the data processing and refinement and helped in the writing, M.B. helped in the design and performed biochemical experiments, J.D.F.-G. designed and supervised the computer simulations and ab initio calculations and helped in the writing, M. Liao supervised the EM studies and helped in the writing. D.M.M. devised and supervised experiments, analyzed data, and helped in writing the paper.

## Competing interests
The authors declare no competing interests.
