## [Peer Review File · Communications Biology]

Reviewers' comments:

Reviewer #1 (Remarks to the Author):

The manuscript by Luo et al describes studies of the inhibitory properties of bedaquiline with respect to mitochondrial ATP synthase. This drug has already been approved for treatment of tuberculosis, as an agent against *Mycobacterium tuberculosis*, but is shown here to be inhibitory towards ATP synthase of yeast and human ATP synthase.

The authors carried out parallel studies comparing the inhibitory properties of bedaquiline and oligomycin (a known, high affinity inhibitor of yeast and human ATP synthase). They used both purified enzymes, and submitochondrial particles to assay rates of ATP hydrolysis. Both inhibitors have sub-micromolar IC₅₀ against the purified yeast enzyme, and ~ 1 micromolar IC₅₀ against the yeast enzyme in submitochondrial particles. Similar potency was demonstrated against ATP synthesis in yeast mitochondria. Inhibition of ATP synthesis by human mitochondria was also demonstrated, using both inhibitors, at ~ micromolar IC₅₀ values.

Binding of bedaquiline was demonstrated by cryo-EM studies of the yeast ATP synthase (F₁F_o). The inhibitor was located at a site similar to where oligomycin binds, along the ring of c subunits, near the interface with subunit a.

Molecular Dynamics simulations showed that protonated bedaquiline could bind to a c-subunit in which the key glutamic acid is deprotonated. Two poses were identified, but only one was feasible in a complex with both a and c subunits.

The authors also examined amino acid sequences of a and c subunits from several relevant species.

The conclusions presented are that the enzyme assays and cryo-EM structures are consistent with a similar mode of binding between mycobacteria and mitochondrial enzymes, and this is supported by the high conservation at the amino acid and structural level. The discrepancy with earlier studies that did not find inhibition of mitochondrial enzymes was suggested to be due differences in assay conditions, in particular lipids, which can greatly affect the apparent affinities of hydrophobic compounds.

The findings appear sound, and are worthy of broad dissemination, given that this is a widely prescribed drug, and that there are known health issues with a small percentage of patients.

1. There was not much discussion of the variation in IC₅₀ values, for example comparing yeast purified enzyme, SMP and mitochondria. Could it result because of endogenous compounds, or does it reflect differences in lipid environment? This seems relevant to consideration of differences with prior studies.

2. No mention was made of how the bedaquiline was handled. What was the stock solution, solvent, etc.

3. Table 1. There are some problems with the units for purified enzyme, and human mitoplasts, where some of the SEM are given in nM, it appears. e.g., 0.027 ± 2.5 . The Table lacks a proper title. It also shows Human data.

4. Supplemental Figures S3 and S4 have some clipping of text, etc, due to overlapping images. The legend to Fig. S3, panel C is confusing, referring to red boxes.

5. Figure S7 shows Subunit c, not b, as the heading indicates.

6. Some references lack publication years. Also, reference 25 does not seem to address inhibition of mitochondrial enzymes. I also noticed that reference 24, which appears to be the primary source of prior analysis of inhibition by bedaquiline of the mitochondrial enzymes is only 3 pages, and lacks basic description of the assays.

7. It would be helpful to include chemical structures of bedaquiline and oligomycin in the beginning of the manuscript.

Minor points, typos, etc.

line 20 (Abstract) suggestion: has been approved
line 45 suggestion: ATP synthase comprises
line 64 conserved in the mammalian enzyme (delete "with")
line 82 here and elsewhere, it is helpful to identify this as yeast (or human)
submitochondrial particles
line 155 suggestion: is confined to subunit-c.
line 189 remarkable
line 192 pair of c-subunits

Reviewer #2 (Remarks to the Author):

Bedaquiline (BDQ), a diarylquinoline, is a new FDA-approved drug that has been used to treat patients infected with antibiotic resistant *Mycobacterium tuberculosis*. BDQ has been shown to inhibit bacterial ATP synthase by bind to the C-ring of Fo and the epsilon subunit of F1. Although BDQ is effective in treatment of tuberculosis, it prolongs the QT interval with an attendant increased risk of fatal arrhythmia. Previous reports indicating that the ATPase activity of human mitochondrial ATP synthase has an IC50 of 200 micromolar for BDQ, which is 20,000 fold higher than of the *Mycobacterium* enzyme, suggesting that the previously noted the side effect of BDQ on cardiac activity were unlikely to be related to effects of the drug on ATP production in humans. In the present study the IC50 of the yeast and human mitochondrial ATPase activities were found to be in the range of 0.7-1 micromolar, which is only 40 times higher than in bacteria. This would indicate that the effect of BDQ on cardiac electrical properties may stem from decreased ATP production by heart muscle mitochondria. The authors speculate that the discrepancy in the inhibition of ATPase activities by BDQ may be explained the different assays used in the two studies. The second important finding to have emerged from this study is the mode of binding of BDQ to the Fo component of the bacterial and mitochondrial ATP synthases. Whereas the BDQ was reported to bind to the C-ring, the structural data presented in this study indicates that the binding involves interaction of the BDQ with amino acid side chains on both the C-ring and subunit A of the mitochondrial enzyme. This in turn suggests that it may be possible to modify the structure of this diarylquinoline in a way that would reduce its binding affinity for the mitochondrial but not bacterial ATP synthase.

This is a careful and well documented study on the inhibitory activity and mode of interaction of an important antibiotic with the mitochondrial ATP synthase. The IC50 of 0.66 micromolar on the ATPase activity of the human enzyme is sufficiently low to explain the observed prolongation of the QT interval as a consequence of BDQ inhibition of the mitochondrial ATP synthase. Similarly, the difference in the mode of binding of BDQ to the bacterial and mitochondrial ATP synthase open the way to selectively decrease it affinity for the human enzyme while retained its inhibitory activity against the bacterial ATP synthase . Overall this is an important contribution that merits publication.

Comments:

1. The explanation offered for the difference in the inhibitory activity of BDQ here on the human mitochondrial ATP synthase and that reported in other studies would be easier to judge if the human mitochondrial and *Mycobacterium* membranes had been assayed under the same conditions. In the absence of this information the authors should at least compare the specific activities of the ATPase in their and other studies.

2. Binding of the BDQ to the interface of the C-ring and subunit A is consistent with the interaction of the inhibitor with the side chains on the two ATP synthase subunits and the altered conformation of the two subunit A alpha helices in the BDQ bound state. This evidence does not entirely exclude the possibility that the conformational changes may result from the displacement of the subunit A induced by binding of BDQ to the C-ring alone. Have the authors considered or discounted this?

Response to Reviewers.

Reviewer 1.

1. There was not much discussion of the variation in IC50 values, for example comparing yeast purified enzyme, SMP and mitochondria. Could it result because of endogenous compounds, or does it reflect differences in lipid environment? This seems relevant to consideration of differences with prior studies.

Response. This is an important comment. We have added to the discussion (lines 213-228). One issue that we did not discuss was the difference between inhibition of ATP synthesis vs. hydrolysis. Since BDQ is bound at the interface of the a/c subunits, in principle, it is possible that rotation of the c-ring could displace the BDQ during ATP synthesis, but not hydrolysis. Rotation in the direction of ATP hydrolysis would be predicted to tighten the interaction of BDQ with the a/c interface by reducing koff. But this is speculation as it would require that the proton pathway is intact in both directions in the ATP synthase – and we really don't know if this is true. But this phenomenon might also partly explain the low IC50 for ATP hydrolysis using isolated yeast F1Fo incorporated into bicelles.

2. No mention was made of how the bedaquiline was handled. What was the stock solution, solvent, etc.

Response. Added (lines 272-282).

3. Table 1. There are some problems with the units for purified enzyme, and human mitoplasts, where some of the SEM are given in nM, it appears. e.g., 0.027 ± 2.5 . The Table lacks a proper title. It also shows Human data.

Response. The title has been modified and the units corrected. Thank you!

4. Supplemental Figures S3 and S4 have some clipping of text, etc, due to overlapping images. The legend to Fig. S3, panel C is confusing, referring to red boxes.

Response. Corrected.

5. Figure S7 shows Subunit c, not b, as the heading indicates.

Response. Corrected.

6. Some references lack publication years. Also, reference 25 does not seem to address inhibition of mitochondrial enzymes. I also noticed that reference 24, which appears to be the primary source of prior analysis of inhibition by bedaquiline of the mitochondrial enzymes is only 3 pages, and lacks basic description of the assays.

Response: We remove reference 25 and added a replacement. These are now references, 10, 21, 24, 25. Reference 24 is still the primary source demonstrating that BDQ does not inhibit the mitochondrial ATP synthase. The other listed references provide indirect evidence. We too are frustrated to the lack of details in the prior publications. We certainly believe that the differences observed are due to differences in the assay.

7. It would be helpful to include chemical structures of bedaquiline and oligomycin in the beginning of the manuscript.

Response: Done. Included in Figure 1.

Minor points, typos, etc.

Response: All the following suggestions were taken.

line 20 (Abstract) suggestion: has been approved

line 45 suggestion: ATP synthase comprises

line 64 conserved in the mammalian enzyme (delete "with")

line 82 here and elsewhere, it is helpful to identify this as yeast (or human) submitochondrial particles

line 155 suggestion: is confined to subunit-c.

line 189 remarkable

line 192 pair of c-subunits

Reviewer 2.

1. The explanation offered for the difference in the inhibitory activity of BDQ here on the human mitochondrial ATP synthase and that reported in other studies would be easier to judge if the human mitochondrial and Mycobacterial membranes had been assayed under the same conditions. In the absence of this information the authors should at least compare the specific activities of the ATPase in their and other studies.

Response: We believe that the reviewer is asking us to compare the specific activities for ATP synthesis for the human mitochondria as determined in the past studies that assayed the inhibition of BDQ.

The specific activity for ATP synthesis for human mitochondria is not reported in ref. 24, but reported by another laboratory using the same methods, but from a different cell line (15 nmoles/min/mg) (ref 45). This is compared to 50 nmoles/min/mg we get for yeast mitochondria and 9 nmoles/min/mg protein for mitochondria from HEK293 cells. Also, we have included the specific activities for ATP synthesis from

membranes derived from *M. bovis* (0.27 nmoles/m/mg) and *M. smegmatis* (0.96 nmoles/min/mg). We have included these in Methods, lines 369-375.

2. Binding of the BDQ to the interface of the C-ring and subunit A is consistent with the interaction of the inhibitor with the side chains on the two ATP synthase subunits and the altered conformation of the two subunit A alpha helices in the BDQ bound state. This evidence does not entirely exclude the possibility that the conformational changes may result from the displacement of the subunit A induced by binding of BDQ to the C-ring alone. Have the authors considered or discounted this?

Response. This is a possibility that we have not excluded, but have no evidence for it based on the cryo-EM density. This is added on lines 211-212. To this point, the movement of the residues in subunit-a in the binding site is consistent with BDQ binding at that site. While it is possible that BDQ binds to the c-ring alone, it appears that BDQ is bound more tightly at the a/c interface.

REVIEWERS' COMMENTS:

Reviewer #1 (Remarks to the Author):

All matters were addressed. No further comments.

Reviewer #2 (Remarks to the Author):

I recommend that this revision be accepted for publication as the authors have satisfactorily answered my comments.